# Surface Flashover Induced by Metal Contaminants Adhered to Tri-Post Epoxy Insulators under Superimposed Direct and Lightning Impulse Voltages

**DOI:** 10.3390/polym14071374

**Published:** 2022-03-28

**Authors:** Qi Hu, Qingmin Li, Zhipeng Liu, Naifan Xue, Hanwen Ren, Manu Haddad

**Affiliations:** 1State key Laboratory of Alternate Electrical Power System with Renewable Energy Sources, North China Electric Power University, Beijing 102206, China; 15650758736@163.com (Q.H.); lzp40@ncepu.edu.cn (Z.L.); nev777@ncepu.edu.cn (N.X.); rhwncepu@ncepu.edu.cn (H.R.); 2Advanced HV Engineering Research Centre, Cardiff University, Cardiff CF24 3AA, UK; haddad@cardiff.ac.uk

**Keywords:** tri-post epoxy insulator, surface charge, metal contaminant, surface flashover, superimposed direct and lightning impulse voltage

## Abstract

Metal contaminants can distort the surface electric field of the tri-post epoxy insulator and cause serious surface charge accumulation, significantly reducing the insulation performance of the insulator under the superimposed DC and lightning impulse voltage. In this paper, an experimental platform for charge accumulation and surface flashover of tri-post epoxy insulators under the superimposed DC and lightning impulse voltage was built, by surface point measurement and charge inversion calculation, the surface charge distribution characteristics of tri-post insulators with attached particles was experimentally explored and the influence law of attached metal particles on the charge accumulation was discussed. The results show that metal particles can cause a surge in the surface charge density of the tri-post epoxy insulator, forming bipolar charge spots whose polarity is opposite to that of the adjacent electrodes. The adsorbed metal dust can cause the polarity reversal of adjacent surface charges, forming a large-area unipolar charge spot. Moreover, the flashover voltage of a tri-post insulator under DC superimposed lightning impulse voltage with a clean surface and attached metal particle was measured, and the synergistic induction mechanism of charge spot accumulation and metal particle discharge on the flashover along the face of the tri-post insulator is thereby revealed. Compared with the clean insulators, the surface flashover voltages of tri-post epoxy insulators with metal contaminants adhered decrease under the superimposed voltages of different polarities, but the decline amplitude is greater under the heteropolar composite voltage. When adhered to the middle of the insulator leg, the distribution range of bipolar charge spots is the widest, and the surface flashover voltage decreases sharply, which can drop by 32% compared with the absence of particles. In addition, when the metal dust adsorbed by the tri-post epoxy insulator has a wide distribution range, the impact of metal dust on the flashover voltage is greater than that of the attached metal particles, and its hazard cannot be ignored. The research results can provide a reference for the insulation test method and optimal design of the DC tri-post epoxy insulator.

## 1. Introduction

Direct current gas-insulated metal-enclosed transmission line (DC GIL) has the advantages of large transmission capacity, low power loss, and high laying flexibility. It can make up for the limitations of traditional power transmission methods and has high application prospects and promotion value [1,2,3]. However, the operational reliability of epoxy insulators faces two key challenges during the service of DC GIL. On one hand, lightning impulse overvoltage may occur on the GIL bus, and the insulator needs to withstand the DC superimposed lightning impulse composite voltage [4]. On the other hand, the gas-solid interface of epoxy insulators accumulates charges and adsorbs metal contaminants under DC voltage, resulting in the electric field increasing at the end of the particle, and transferring the area where the electric field intensity is most concentrated from the triple point to the tips of the metal particle. As the particle gets closer to the conductor, the electric field increases significantly. When the particle is attached to the junction of the insulator and conductor, the decrease of insulating strength can be up to 60% compared to the clean setup [5,6]. When the insulators face the above two challenges at the same time, the surface insulation performance can decrease significantly, which is prone to causing the surface flashover.

To solve the above problems, the researchers studied the insulation performance of insulators under DC superimposed lightning impulse composite voltage. For clean insulators, T. Hasegawa has found that a large number of surface charges accumulated on the insulator under DC voltage, and the flashover voltage of the insulator under DC superimposed lightning impulse voltage was closely related to the pre-stressed time of DC voltage [7,8]. H. Fujinami applied DC voltage to the single-post insulator model for a certain period, and then applied an impulse voltage with the opposite polarity to the DC voltage. The results showed that the insulator flashover voltage decreased with the increase of surface charge density and presented the same trend under different gas pressures [9]. G. Ma studied the effect of temperature on the surface flashover of basin-type insulators under DC superimposed lightning impulse voltage, and found that with the increase of bus temperature, the change rate of insulator flashover voltage increased [10]. When metal contaminants adhere to the surface of the insulator, the surface electric field of the insulator would be distorted, resulting in a more serious surface charge accumulation and changing the flashover characteristics of the insulator under the superimposed DC impulse voltage [11,12]. A. Bargigia found that the flashover voltage of the insulators with linear metal particles adhered decreased compared with that of clean insulators. When the polarities of the DC voltage and the impulse voltage were opposite, the flashover voltage decreased more significantly [13]. S. Tenbohlen found that surface charge would affect the streamer initiation process of insulators with particles adhered. When negative charges existed on the insulator surface, the electron avalanche would be promoted under the positive impulse voltage, thereby reducing the insulator streamer initiation voltage [14]. 

However, previous research mainly focused on the insulating performance of basin and single-post insulators. To compensate for the thermal extension and mechanical strain of the conductors, in addition to the basin insulators, tri-post epoxy insulators are widely used in the GIL [15,16]. The shape of the insulator has a great influence on the electric field distribution, charge accumulation, and creeping discharge process [17,18]. The existing research cannot analyze the insulation characteristics of the tri-post epoxy insulator under the DC superimposed lightning impulse composite voltage. Moreover, the size of metal contaminants affects the insulation performance of epoxy insulators. The existing research mostly uses linear metal particles above millimeters to carry out experiments, without considering the influence of metal dust on insulators.

To this end, an experimental platform for charge accumulation and surface flashover of the tri-post epoxy insulator was built. Further, by means of the surface potential measurement and surface charge inversion algorithm, we experimentally explored the surface charge distribution characteristics of tri-post insulators with attached particles, on this basis, the influence law of attached metal particles on the charge accumulation was discussed. Finally, we measured the flashover voltage of a tri-post insulator under DC superimposed lightning impulse voltage with a clean surface and attached metal particle, and the synergistic induction mechanism of charge spot accumulation and metal particle discharge on the flashover along the face of the tri-post insulator is thereby revealed, which provided a reference for the insulation test method and optimal design of the DC GIL tri-post epoxy insulator.

## 2. Experimental Setup

### 2.1. Experimental Platform

To obtain the surface flashover characteristics of the tri-post epoxy insulator under the DC superimposed lightning impulse voltage, an experimental platform for the tri-post epoxy insulator was established as shown in Figure 1, including a superimposed voltage power supply, experimental chamber, surface charge measuring device, and oil bath circulating heating device, etc.

The superimposed voltage power supply was composed of a DC power generator and an impulse generator, which was connected to the high-voltage bushing of the experimental chamber and could apply a composite voltage to the test insulator. To explore the effect of the composite voltage of different polarity combinations, experiments were carried out under positive DC superimposed positive impulse and positive DC superimposed negative impulse respectively. In addition, to avoid mutual influence between the two power sources, the DC power generator and the impulse generator were connected in series with a 10 MΩ protective resistor and a 10 nF DC blocking capacitor, respectively. The surface charge measuring device included an electrostatic voltmeter Trek 341B, electrostatic probe Trek 3455ET, a five-axis manipulator arm, multi-axis motion controller, etc. The multi-axis motion controller can control the manipulator arm to drive the electrostatic probe to scan and test the surface potential of the insulator. Then, the surface charge distribution of the tri-post epoxy insulator could be calculated by the surface charge inversion algorithm [19]. The oil bath circulating heating device was composed of a modified high-voltage conductive pole, storage tank, and high-temperature oil pump. The temperature of the conductive pole is raised to 333 K by the heat exchange with high-temperature oil, simulating the current-carrying heating of the conductive pole in the operating environment [10]. The experimental insulators were made of Al_2_O_3_-filled epoxy resin, and the outer and inner diameters were 200 mm and 60 mm, respectively. They were manufactured by Taikai Group Co., Ltd., Taian, China and installed in a coaxial cylindrical electrode structure as GIL.

### 2.2. Test Method

Before the test, to eliminate the interference of the residual charge, the insulator was wiped with absolute ethanol and dried in a drying oven for 12 hours. Considering the type and size of metal contaminants in GIL, larger-sized linear metal particles (0.2 mm in diameter and 10 mm in length) and micron-sized metal dust (approximately 15 μm in diameter) were selected for experiments [20]. The metal particles used in the experiment were linear aluminum wires, which were adhered to different positions of the insulator by conductive silver glue (the particles were on the intersection line with the smallest creepage distance, and A, B, and C were, respectively, 5, 25, and 45 mm from the ground electrode), as shown in Figure 2. The metal dust used in the experiment was aluminum dust, which was weighed to 0.3 g by a high-precision balance and placed on the inner surface of the grounded enclosure (D, 50 mm away from the insulator). Under the influence of the electric field, a part of the metal dust would be adsorbed to the surface of the tri-post epoxy insulator, thus affecting the surface charge accumulation and surface flashover process of the insulator. After the metal contaminants are set, the experimental chamber was evacuated and filled with SF_6_ gas to 0.1 Mpa, and then the oil bath circulating heating device was turned on and heated continuously for 5 hours to ensure that the temperature of the conductive pole reached 333 K.

During the test, the DC power generator applied +100 kV DC voltage to the test insulator until the surface charge accumulation on the insulator approaches saturation. Pre-experimental results showed that when the temperature of the conductive pole was 333 K, the saturation time of the surface charge accumulation process of the insulator did not exceed 5 hours. Therefore, the DC power generator was turned off after continuously applying the DC voltage for 5 hours, and the surface charge distribution of the insulator was measured by the surface charge measuring device. Then, the above steps were repeated. The DC power generator was maintained after continuously applying the DC voltage for 5 hours. The impulse generator was turned on and the standard lightning impulse voltage (1.2/50 μs) was superimposed to the tri-post epoxy insulator by the step method. Each set of experiments under the same conditions was repeated five times.

## 3. Surface Charge Accumulation Characteristics of Tri-Post Epoxy Insulators under DC Voltage

### 3.1. Surface Charge Distribution of Tri-Post Epoxy Insulator with Adhered Metal Contaminants

When the temperature of the conductive pole is 333 K and the +100 kV DC voltage is applied for 5 hours, the measured surface charge distribution of the tri-post epoxy insulator is shown in Figure 3. Without metal particles adhered, the bottom area of the tri-post epoxy insulator legs accumulates negative charges, the rest area of the legs accumulates positive charges, and the positive charges are most concentrated in the middle area of the legs. When metal particles are attached, the surface charge of the tri-post epoxy insulator surges and forms bipolar charge spots. The polarity of the charge spots is opposite to that of the adjacent electrode. The top area of the leg near the high-voltage electrode accumulates a negative charge spot, and the bottom area of the leg near the grounded electrode accumulates a positive charge spot. In addition, the attachment position of metal particles can affect the distribution range of charge spots. The charge spots are most widely distributed when the particles are attached to B in the middle of the insulator leg.

To further study the influence of metal particles on the surface charge accumulation process of the tri-post epoxy insulator, the surface charge density on the intersection line where metal particles are attached is selected for further analysis, and the results are shown in Figure 4. The black and red lines represent the surface charge density in the presence or absence of metal particles, respectively. The shaded areas in pink and blue represent the positive and negative charge accumulation caused by metal particles, respectively. In each set of experiments, the metal particle causes the positive charge accumulation at the leg bottom area and the negative charge accumulation at the middle and top areas of the leg.

When metal dust is near the tri-post epoxy insulator, the metal dust is mainly adsorbed in the middle and top areas of the legs of the tri-post epoxy insulator, as shown in Figure 5a. The attached metal dust causes the polarity of the adjacent surface charges to change from positive to negative, forming a large area of negative charge spots on the leg surface of the tri-post epoxy insulator, as shown in Figure 5b.

### 3.2. Influence of Metal Contaminants on the Surface Charge Accumulation of the Tri-Post Epoxy Insulator

There are three main sources of insulator surface charge: bulk conduction current ***j***_v_, surface conduction current ***j***_s_, and gas conduction current ***j***_g_ [21]. Based on the current continuity equation, the surface charge accumulation transient equation can be expressed as: (1)$$\frac{\partial \sigma }{\partial t}=n\cdot j_{v}-n\cdot j_{g}-\nabla \cdot j_{s}$$
where *σ* is the surface charge density, *t* is the time, and ***n*** is the normal vector from the insulator to the gas.

When the insulator is fully dry and clean, the surface charge accumulation equation can be simplified as: (2)$$\frac{\partial \sigma }{\partial t}=\gamma_{v}\cdot E_{\mathrm{Vn}}-\gamma_{g}\cdot E_{\mathrm{Gn}}-\nabla \cdot \left(\gamma_{s}\cdot E_{s}\right)$$
where *γ*_v_, *γ*_s_, and *γ*_g_ are the bulk conductivity of the insulator, the surface conductivity and the gas conductivity, respectively; *E*_Vn_ and *E*_Gn_ are the normal electric field strength on the insulator side and the gas side, respectively; ***E***_s_ is the surface tangential electric field strength of the insulator.

The surface conductivity *γ*_s_ of the insulator material is much smaller than the bulk conductivity *γ*_v_. The surface conduction current is very small, so the charge moving along the surface of the insulator is negligible [21]. The surface charge accumulation of the insulator is mainly related to the current difference on both sides of the solid-gas interface. When the bulk conduction current ***j***_v_ dominates, positive charges accumulate in the middle area of the tri-post epoxy insulator leg; when the gas conduction current ***j***_g_ dominates, negative charges accumulate in the middle area of the tri-post epoxy insulator leg. The experimental results in Figure 3 show that positive charges accumulate in the middle area of the legs of the tri-post epoxy insulator without metal contaminants adhered. Therefore, it is inferred that the bulk conduction current of the insulator plays a dominant role in the surface charge accumulation process of the clean tri-post epoxy insulator, as shown in Figure 6.

When metal contaminants are attached to the surface of the tri-post epoxy insulator, negative charges accumulate in the middle area of the legs. It can be concluded that the gas conduction current increases significantly in the presence of metal contaminants and plays a dominant role in the charge accumulation process, as shown in Figure 7. The tips of metal particles can severely distort the surrounding electric field and cause corona discharges. The charged ions generated by the corona discharge will migrate along the electric field lines in the insulating gas, and some of the charged ions migrate and deposit on the surface of the insulator, thereby changing the surface charge distribution of the insulator. When positive DC voltage is applied to the high-voltage electrode, the electric field lines in the top and middle regions of the tri-post epoxy insulator leg are oriented to the gas, so the negative ions migrate to the surface of the insulator and form a negative charge spot. The electric field lines in the bottom region of the legs are oriented to the insulator, so the positive ions migrate to the surface of the insulator and form a positive charge spot. Compared with other positions, when the metal particle is attached at B in the middle of the leg, the electric field lines near the particles all pass through the surface of the insulator. The charged ions generated by corona discharge are more inclined to migrate to the surface of the insulator along the electric field lines, so the distribution of the charge spot is wider. Similarly, when metal dust is adsorbed on the surface of the tri-post epoxy insulator, the partial discharge between the dust particles can also generate charged ions, resulting in an increase in the gas conduction current and the accumulation of charge spots on the surface of the insulator. Due to the low intensity of partial discharge caused by metal dust, the charged ions generated by the discharge are mainly deposited on the insulating surface near the dust. Therefore, only negative charge spots are accumulated on the tri-post epoxy insulator leg attached to the metal dust.

## 4. Surface Flashover Characteristics of Tri-Post Epoxy Insulators under Superimposed DC and Impulse Voltage

### 4.1. Surface Flashover Characteristics of Tri-Post Epoxy Insulators in Clean State

On the basis of mastering the surface charge distribution of the tri-post epoxy insulator under DC voltage, the impulse voltage is applied to the insulator with saturated charge accumulation, and the surface flashover characteristics of the tri-post epoxy insulator under the superimposed DC impulse voltage is studied. Firstly, the effect of pre-stressed DC voltage on flashover voltage is studied without metal contamination. +100 kV positive DC voltage *U*_DC_ is applied to the insulator for 0 and 5 hours respectively, and then a certain amplitude of positive or negative standard lightning impulse voltage *U*_LI_ was superimposed to obtain the surface flashover voltage *U*_s_. The surface flashover amplitude calculation method is shown in Figure 8, where a value greater than 0 represents positive polarity and a value less than 0 represents negative polarity.

The relationship between the flashover voltage of the tri-post epoxy insulator and pre-stressed time is shown in Figure 9. Under the positive DC superimposed positive impulse composite voltage, the average flashover voltage after pre-stressed +100 kV DC voltage for 0 and 5 hours is 302 kV and 290 kV, respectively. The average flashover voltage amplitude decreased by 4% due to the accumulated charge on the insulator surface. Under the positive DC superimposed negative impulse composite voltage, the average flashover voltage after pre-stressed +100 kV DC voltage for 0 and 5 hours is –297 kV and –266 kV, respectively. The average flashover voltage amplitude decreased by 10% due to the accumulated charge on the insulator surface. When the surface of the tri-post epoxy insulator is in a clean state, the pre-stressed DC voltage can lead to the decline of flashover voltage, and rates goes down most sharply for the heteropolar composite voltage.

The above phenomenon of the difference in flashover voltage variation under different compound voltages can be explained from the surface charge distribution of the insulator, as shown in Figure 10. Under the positive DC voltage, positive charges are accumulated at the top and middle areas of the tri-post epoxy insulator legs, and negative charges are accumulated at the bottom area of the legs, forming a surface charge electric field *E*_q_. When positive impulse voltage is superimposed, the direction of *E*_q_ is opposite to the applied electric field *E* at the triple point of "electrode-insulator-gas"; when negative impulse voltage is superimposed, the direction of *E*_q_ is the same as the applied electric field at the triple point. The original electric field strength at the triple point of the insulator is larger. Under the heteropolar composite voltage, the surface charge electric field *E*_q_ further enhances the electric field strength of the triple point, which is conducive to the development of discharge and reduces the surface flashover voltage.

### 4.2. Surface Flashover Characteristics of Tri-Post Epoxy Insulators with Adhered Metal Contaminants

+100 kV DC voltage is pre-stressed to the tri-post epoxy insulator with metal contaminants adhered for 5 h, and then the standard lightning impulse voltage is superimposed to obtain the surface flashover voltage. Figure 11 shows the surface flashover path of the tri-post epoxy insulator with metal particles attached. In each set of experiments, the surface flashover path of the tri-post epoxy insulator always passes through the metal particle and develops along the intersection line with the smallest creepage distance.

The relationship between the flashover voltage of the tri-post epoxy insulator and the position of metal contaminants is shown in Figure 12. Under the positive DC superimposed positive impulse composite voltage, the average flashover voltage of the insulators with metal particles attached to A, B, and C is 280 kV, 241 kV, and 232 kV, respectively, and the average surface flashover voltage of the insulators absorbing metal dust is 174 kV. Under the positive DC superimposed negative impulse composite voltage, the average flashover voltage of the insulators with metal particles attached to A, B, and C is –246 kV, –180 kV, and –207 kV, respectively, and the average surface flashover voltage of the insulators absorbing metal dust is –166 kV.

Compared with the insulator in the clean state, the surface flashover voltage of the tri-post epoxy insulator with metal contaminants adhered decreases. In terms of the type of composite voltage, when the metal particle is attached to the insulator surface, the flashover voltage goes down more sharply for the heteropolar composite voltage. In terms of the attachment position of the metal particle, the metal particle at A has the least influence on the surface flashover voltage. When the metal particle is attached to B, the surface flashover voltage decreases sharply, and the average flashover voltage amplitude decreases by 32% under the positive DC superimposed negative impulse composite voltage. In terms of the type of metal contaminants, when the metal dust adsorbed by the tri-post epoxy insulator has a wide distribution range, the metal dust will significantly reduce the surface flashover voltage, and the impact on the insulation performance of the insulator is even greater than that of the larger-sized metal particle.

Figure 13 shows the flashover process of the insulator under the heteropolar composite voltage, with the particle attached at B as an example. Under the positive DC voltage, the electric field is distorted at the ends of the attached particle, triggering corona discharge. The charged ions generated by the corona discharge migrate along the electric field and are deposited on the insulator surface, leading to the accumulation of bipolar charge spots, as shown in the corona discharge and charge accumulation stage of Figure 1. When lightning impulse voltage is higher, the corona discharge is intensified at the ends of the particle, and the collision ionization is thereby strengthened in the gas. So that the discharge at both ends of the particle develops toward the conductor and enclosure and forms the streamer channel as shown in the streamer stage of Figure 13a. 

As the streamer channel develops to the electrode, the discharge promotes the development of the streamer on the other side of the particle, eventually resulting in flashover along the insulator surface. As shown in Figure 13a,b, when superimposed negative lightning impulse, *E*q enhances the electric field at the particle ends, further intensifying the corona discharge. In addition, the charge accumulation on the insulator surface will strengthen the electric field at the head of the streamer, increasing the amount of charge inside the streamer, as well as promoting the further development of the streamer channel. As a result, the flashover along the surface is more likely to occur under heteropolar composite voltage.

Besides, when the metal particle is attached to the middle area of the insulator leg, the distribution range of the bipolar charge spots is wider, and the charge accumulation promotes the discharge development at both ends of the particles, so the flashover voltage decreases most severely. When attached to the bottom or top area of the insulator leg, the particle is completely in the "charge spot accumulation area". Thus, the surface charge promotes the development of discharge at one end of the particles while inhibiting the development of discharge at the other end, making the effect of particle attachment on the flashover voltage relatively small. In particular, the electric field strength on the insulator surface near the conductor is higher and the streamer channel is more likely to form, therefore the influence of particles attached to the top area of the insulator leg is greater than that of the bottom.

Compared with metal particles of larger sizes, metal dust has less distortion to the sur-rounding electric field. However, when the metal dust adsorbed by the tri-post epoxy insulator has a wide distribution range, the impact of metal dust on the flashover voltage is greater. The possible reason is that partial discharge occurs between metal dust, forming a local short-circuit area. When the middle and top areas of the insulator legs are covered with metal dust, a large short-circuit area is formed on the surface of the insulator, which greatly shortens the creepage insulation distance of the insulator, so the surface flashover is more likely to occur.

With particles attached to the insulator surface, conditioning with AC voltage does not change the flashover behavior for stress with lightning impulses [22], while the flashover voltage under DC superimpose lightning impulses is lower than that under lightning impulses alone. In terms of the effect of particle location on flashover voltage, the decrease in flashover voltage is particularly significant when particles are connected to the conductor under AC superimpose lightning impulses [23]. However, under DC superimpose lightning impulses, when the metal particle is attached to the middle area of the insulator leg, the flashover voltage decreases most severely. For this reason, the charge accumulation on the insulator surface is weak at AC, which has almost no effect on flashover of lightning impulses. When the particles are connected to the electrodes, the distortion of the electric field strength becomes the strongest, forming a stable partial discharge channel and thus the flashover voltage is significant reduction under AC superimpose lightning impulses. Whereas, under DC voltage, charge accumulation is formed at the end of the particles, and the particles have the maximized coupling effect on the field strength formed by the charges when particle located in the middle area of the insulator, the flashover voltage decreases most severely.

## 5. Conclusion

In this paper, an experimental platform for charge accumulation and surface flashover of the tri-post epoxy insulator was built, and the surface charge distribution of the tri-post epoxy insulator with metal contaminants adhered was obtained. Furthermore, the influence of the metal contaminants on the flashover of the tri-post epoxy insulator under the DC superimposed lightning impulse voltage was studied, the main conclusions are summarized as follows:(1)Under the DC voltage, compared to the clean surface, the metal particle attached to the insulator leads to the electric field strength being enhanced and partial discharge at the attachment location, causing a surge in the surface charge density of the tri-post epoxy insulator, forming bipolar charge spots whose polarity is opposite to that of the adjacent electrodes. The adsorbed metal dust can cause the polarity reversal of adjacent surface charges, forming a large-area unipolar charge spot whose polarity is opposite to that of the high-voltage electrode.(2)When the metal particle is attached to the insulator, the region with the most concentrated electric field is shifted from the triple bonding point to the tip of the particle. Under the superimposed voltages, the distortion of the electric field at the end of the particle and the electric field formed by the surface charge makes the insulator streamer onset voltage decrease, so that the surface flashover voltages of tri-post epoxy insulators with particle adhered decrease, compared with the clean insulators. Under the heteropolar composite voltage, the surface charge further enhances the electric field strength of the particle tips and promotes the development of an electron avalanche. Therefore, the flashover voltage decreases more sharply for the heteropolar composite voltage.(3)The influence of the metal particle on the flashover voltage of the tri-post epoxy insulator is closely related to the attachment positions. When the metal particle is attached to the middle area of the insulator leg, the distribution range of the bipolar charge spots is wider. Therefore, the charge spot has a stronger promoting effect on surface discharge, and the insulator flashover voltage drops sharply, which can be reduced by 32% compared with the absence of particles. In addition, when the metal dust adsorbed by the tri-post epoxy insulator has a wide distribution range, the impact of metal dust on the flashover voltage is greater than that of the attached metal particle, and its hazard cannot be ignored.(4)The proposed flashover characteristics of tri-post insulators under the combined effect of charge accumulation and metal particles can provide guidance for the optimal design of DC GIL tri-post insulators. The authors will in future research on the inhibition of particle adsorption and charge accumulation by insulator surface coatings.

## Figures and Tables

**Figure 1 polymers-14-01374-f001:**
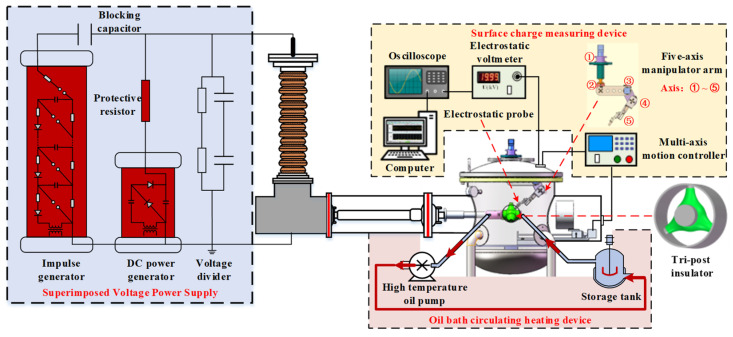
Experimental platform for charge accumulation and surface flashover of tri-post epoxy insulators.

**Figure 2 polymers-14-01374-f002:**
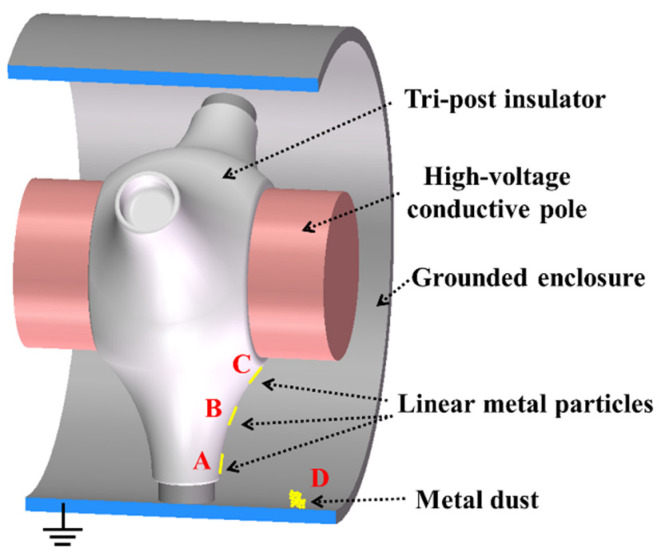
Locations of metal contaminants.

**Figure 3 polymers-14-01374-f003:**
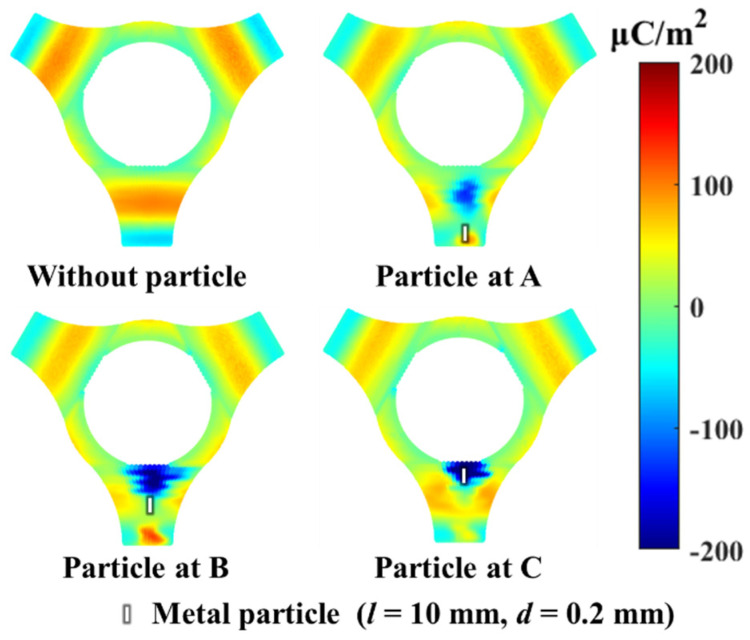
Surface charge distribution of tri–post epoxy insulator with metal particle attached under +100 kV DC voltage.

**Figure 4 polymers-14-01374-f004:**
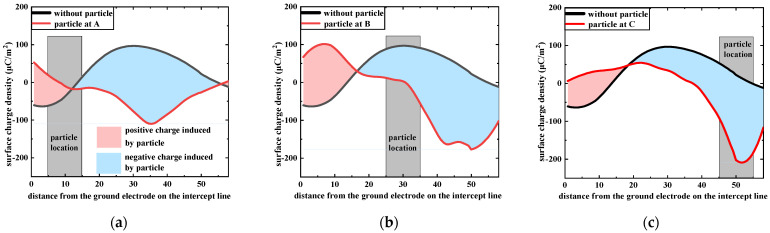
Surface charge distribution on intersection line of tri–post epoxy insulator with metal particle attached. (**a**) Particle at A; (**b**) Particle at B; (**c**) Particle at C.

**Figure 5 polymers-14-01374-f005:**
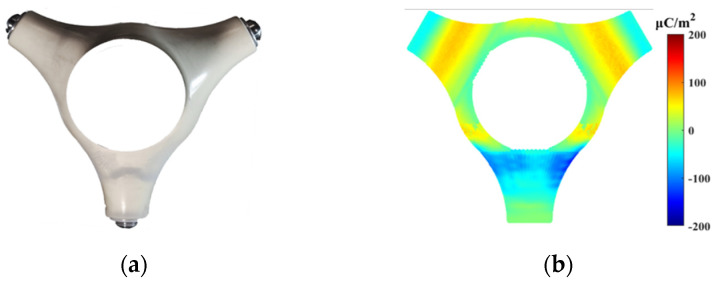
Effect of metal dust on surface charge of tri–post epoxy insulator under +100 kV DC voltage. (**a**) Adsorption distribution of metal dust; (**b**) Charge distribution of metal dust.

**Figure 6 polymers-14-01374-f006:**
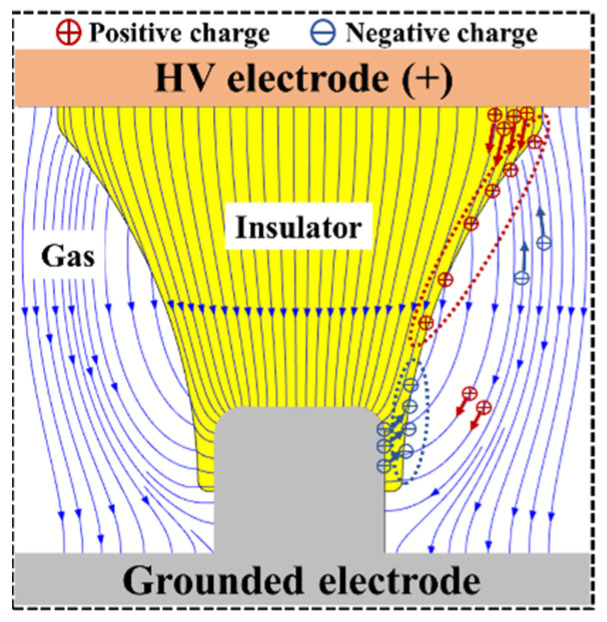
Surface charge accumulation process of tri–post epoxy insulator in clean state.

**Figure 7 polymers-14-01374-f007:**
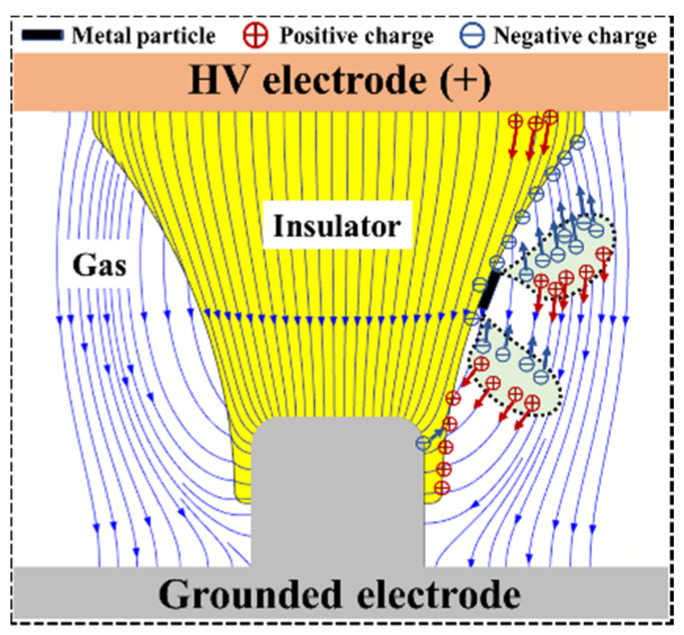
Surface charge accumulation process of tri–post epoxy insulator with metal particle attached.

**Figure 8 polymers-14-01374-f008:**
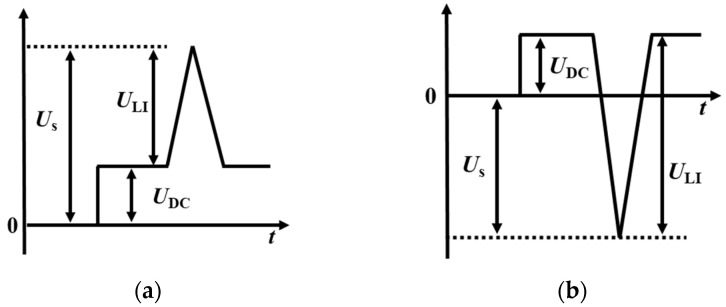
Superimposed DC and lightning impulse voltage waveforms. (**a**) Positive DC + Positive LI; (**b**) Positive DC + Negative LI.

**Figure 9 polymers-14-01374-f009:**
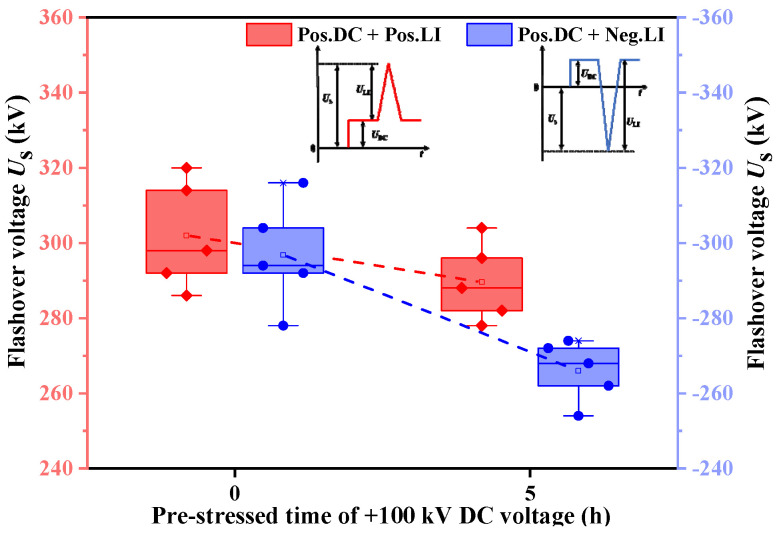
Relationship between flashover voltage of tri–post epoxy insulator and pre–stressed time of +100 kV DC voltage.

**Figure 10 polymers-14-01374-f010:**
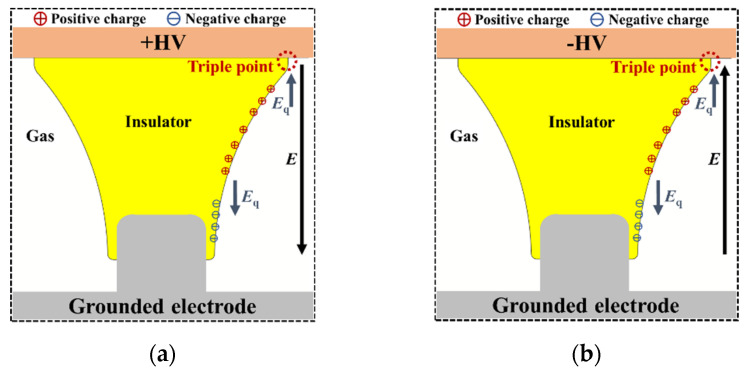
Effect of surface charge accumulation on surface flashover of tri–post epoxy insulator. (**a**) Positive DC + Positive LI; (**b**) Positive DC + Negative LI.

**Figure 11 polymers-14-01374-f011:**
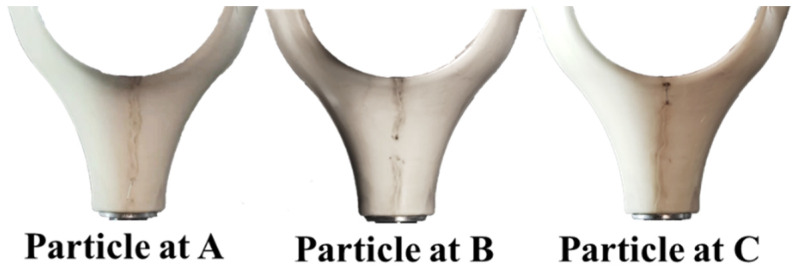
Surface flashover path of tri-post epoxy insulator with metal particles attached.

**Figure 12 polymers-14-01374-f012:**
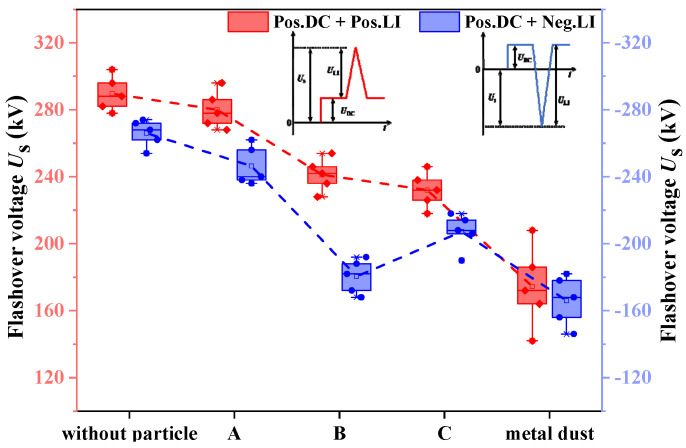
Relationship between flashover voltage of tri–post epoxy insulator and the position of metal contaminants.

**Figure 13 polymers-14-01374-f013:**
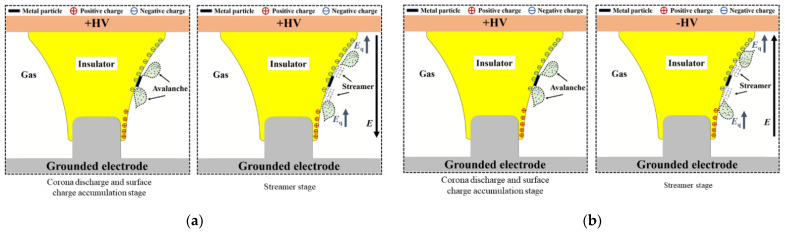
Flashover process of tri–post insulator with attached metal particle under superimposed DC and impulse voltage. (**a**) Positive DC + Positive LI; (**b**) Positive DC + Negative LI.

## Data Availability

The data presented in this study are available on request from the corresponding author.

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
