# Peer review of "Surface Flashover Induced by Metal Contaminants Adhered to Tri-Post Epoxy Insulators under Superimposed Direct and Lightning Impulse Voltages"

_polymers, 2022, doi:10.3390/polym14071374_

Round 1

Reviewer 1 Report

Dear Authors,

Thank you very much for the interesting paper.

The paper presents a study about surficial flashover caused by metal contaminants adhered to tri-post epoxy insulators under DC and lightning impulse voltages. Authors say, that metal parts may distort the surface electric field of epoxy insulator and cause surface charge accumulation, what decreases DC and lightning impulse breakdown voltage. Their results showed that metal particles can cause a surge in the surface charge density. The research indicates that they are able to provide a reference for the insulation test method and optimal design of the DC tri-post epoxy insulator.

Comments and questions:

  1. Introduction chapter well describes the history of study of contaminants presence on epoxy insulator surface, mostly in case of GIL. Also, negative impacts of the presence is well explained.
  2. I think, Introduction chapter should present also the philosophy of negative impact of metal contaminants on epoxy surface. What I mean: how does the metal particle change electric field stress? Where do we observe the increase and reduce the field on the epoxy surface? What is the effect of mentioned increase? Answer my question would help the authors.
  3. Authors results were presented many years ago by prof. Kurt Feser and Mark Fieger from Stuttgart University, Germany, in 1999, almost 20 years ago. Prof. Kurt Feser and Mark Fieger analyzed also orientation and location of metal particles in GIS / GIL. They study GIS / GIL in case of AC, not DC and pulse voltage. Anyway, I would expect to compare authors results to Feser and Fieger results.
  4. Obtained results are very interesting, but the explanation of the results is not complete. I would expect more explanation, especially physical explanation, of presented results.
  5. It is the same comments in Conclusions, where authors compare contaminated and clean surface. Please explain what is really difference between mentioned cases from electric field point of view. It corresponds to comment number 2 with my questions.

Reviewer 2 Report

Abstract should be given as more interesting. Express at least one of the main aspects and features of the paper.

At the end of Introduction section, it would be better to add the paper's organization in different paragraph.

Improve the conclusion.

Manuscript must be presented in highlight the novelty, contribution, and applicability of the work.

Future work is missing.

Please check the manuscript for wrong choice of words, grammatical errors and incoherent sentence structure

Round 2

Reviewer 1 Report

All my comments and suggestions are included in the paper by authors. So, in my opinion, the paper is ready to be published in present form.

Reviewer 2 Report

Accept the manuscript for publication in its present form.